# Learning Coefficient of Vandermonde Matrix-Type Singularities in Model Selection

**DOI:** 10.3390/e21060561

**Published:** 2019-06-04

**Authors:** Miki Aoyagi

**Affiliations:** Department of Mathematics, College of Science & Technology, Nihon University, 1-8-14, Surugadai, Kanda, Chiyoda-ku, Tokyo 101-8308, Japan; aoyagi.miki@math.cst.nihon-u.ac.jp; Tel.: +81-3-3259-0883

**Keywords:** learning coefficient, Kullback function, singular learning machine, resolution of singularities

## Abstract

In recent years, selecting appropriate learning models has become more important with the increased need to analyze learning systems, and many model selection methods have been developed. The learning coefficient in Bayesian estimation, which serves to measure the learning efficiency in singular learning models, has an important role in several information criteria. The learning coefficient in regular models is known as the dimension of the parameter space over two, while that in singular models is smaller and varies in learning models. The learning coefficient is known mathematically as the log canonical threshold. In this paper, we provide a new rational blowing-up method for obtaining these coefficients. In the application to Vandermonde matrix-type singularities, we show the efficiency of such methods.

## 1. Introduction

In recent studies, real data associated with, for example, image or speech recognition, psychology, and economics, have been analyzed by learning systems. Hence, many learning models have been proposed, and thus, the need for appropriate model selection methods has increased.

In this section, we first introduce the widely-applicable information criterion (WAIC) [1,2,3,4,5,6,7] and cross-validation in Bayesian model selection.

Let q(x) be a true probability density function of variables, $x\in R^{N}$, and let $x^{n}:={\left\{x_{i}\right\}}_{i=1}^{n}$ be *n* training samples selected from q(x) independently and identically. Consider a learning model that is written in probabilistic form as p(x|w), where $w\in W\subset R^{d}$ is a parameter.

Suppose that the purpose of the learning system is to estimate the unknown true density function q(x) from $x^{n}$ using p(x|w) in Bayesian estimation. Let ψ(w) be an a priori probability density function on parameter set *W* and $p(w|x^{n})$ be the a posteriori probability density function:
$$p\left(w|x^{n}\right)=\frac{1}{Z_{n}\left(\beta \right)}\psi \left(w\right)\prod_{i=1}^{n}p{\left(x_{i}|w\right)}^{\beta },$$where:
$$Z_{n}\left(\beta \right)=\int_{W}\psi \left(w\right)\prod_{i=1}^{n}p{\left(x_{i}|w\right)}^{\beta }dw,$$for inverse temperature β. We typically set β=1. Define:
$$E_{w}^{\beta }\left[f\left(w\right)\right]=\frac{\int dwf\left(w\right)\psi \left(w\right)\prod_{i=1}^{n}p{\left(x_{i}|w\right)}^{\beta }}{\int dw\psi \left(w\right)\prod_{i=1}^{n}p{\left(x_{i}|w\right)}^{\beta }},$$and:
$$V_{w}^{\beta }\left[f\left(w\right)\right]=E_{w}^{\beta }\left[f{\left(w\right)}^{2}\right]-E_{w}^{\beta }{\left[f\left(w\right)\right]}^{2}.$$

We then have predictive density function $p\left(x|x^{n}\right)=E_{w}^{\beta }\left[p\left(x|w\right)\right]$, which is the average inference of the Bayesian density function.

We next introduce Kullback function K(q||p) and empirical Kullback function $K_{n}\left(q||p\right)$ for density functions p(x),q(x):
$$K\left(q||p\right)=\int q\left(x\right)\log\frac{q(x)}{p(x)}dx,$$
$$K_{n}\left(q||p\right)=\frac{1}{n}\sum_{i=1}^{n}\log\frac{q(x_{i})}{p(x_{i})}.$$

Function K(p||q), which always has a non-negative value and satisfies K(q||p)=0, if and only if q(x)=p(x), is a pseudo-distance between density functions p(x),q(x). We define Bayes training loss $T_{n}$ and Bayes generalization loss $G_{n}$ as follows:
$$T_{n}=-\frac{1}{n}\sum_{i=1}^{n}\log p\left(x_{i}|x^{n}\right),$$and:
$$G_{n}=-\int q\left(x\right)\log p\left(x|x^{n}\right)dx.$$

Additionally, we define Bayesian generalization error $B_{g}$ and Bayesian training error $B_{t}$ as follows:$$B_{g}=K\left(q\left(x\right)∥p\left(x|x^{n}\right)\right)$$and
$$B_{t}=K_{n}\left(q\left(x\right)∥p\left(x|x^{n}\right)\right).$$

Then, we have:
$$\begin{array}{ccc} E[T_{n}] & = & G_{n},\\ B_{g} & = & G_{n}-S,\\ B_{t} & = & T_{n}-S_{n} \end{array}$$for average entropy S=−∫q(x)logq(x)dx and empirical entropy $S_{n}=-\frac{1}{n}\sum_{i=1}^{n}\log q\left(x_{i}\right)$ of the true density function. Value $B_{g}$ describes how precisely the predictive function approximates the true density function.

We define $x^{n}\backslash x_{i}=\left\{x_{1},\ldots ,x_{i-1},x_{i+1},\ldots ,x_{n}\right\}$. The WAIC is denoted by:
$$W_{n}=T_{n}+\frac{\beta }{n}\sum_{i=1}^{n}V_{w}^{\beta }\left[\log p\left(x_{i}|w\right)\right]$$and the cross-validation loss is denoted by:
$$C_{n}=-\frac{1}{n}\sum_{i=1}^{n}\log p(x_{i}|x^{n}\backslash x_{i})$$for n≥2.

Watanabe [2,3,6,7] proved the following four relations:
$$\begin{array}{ccc} E[G_{n}] & = & L\left(w_{0}\right)+\frac{1}{n\beta }\left(\lambda +\frac{\beta -1}{\beta }\nu \right)+o\left(\frac{1}{n\beta }\right),\\ E[T_{n}] & = & L\left(w_{0}\right)+\frac{1}{n\beta }\left(\lambda -\frac{\beta +1}{\beta }\nu \right)+o\left(\frac{1}{n\beta }\right),\\ E[W_{n}] & = & L\left(w_{0}\right)+\frac{1}{n\beta }\left(\lambda +\frac{\beta -1}{\beta }\nu \right)+o\left(\frac{1}{n\beta }\right),\\ E[C_{n}] & = & L\left(w_{0}\right)+\frac{1}{n\beta }\left(\lambda +\frac{\beta -1}{\beta }\nu \right)+o\left(\frac{1}{n\beta }\right) \end{array}$$for learning coefficient λ∈Q and singular fluctuation ν∈R, where $L\left(w\right)=-E_{x}\left[\log p\left(x|w\right)\right]$ and $w_{0}\in W_{0}=\left\{w_{0}\in W∥L\left(w_{0}\right)=\min_{w\in W}L\left(w\right)\right\}$. For singular models, the set $W_{0}$ is typically not a singleton set. Nevertheless, the WAIC and the cross-validation can estimate the Bayesian generalization error without any knowledge of the true probability density function.

These values are calculated from training samples $x_{i}$ using learning model *p*. In real applications or experiments, we typically do not know the true distribution, but only the values of the training errors. Our purpose is to show that these methods are effective. We can select a suitable model from several statistical models by observing these values.

In this paper, we consider the value λ, which is equal to the log canonical threshold introduced in Definition 1. This coefficient is not needed to evaluate the WAIC and the cross-validation in practice, while the learning coefficients from our recent results have been used very effectively by Drton and Plummer [8] for model selection using a method called the “singular Bayesian information criterion (sBIC)”. The method sBIC even works using the bounds of learning coefficients.

It is known that λ=ν=d/2 holds, where *d* is the dimension of the parameter space for regular models. Value λ is obtained by a blowing-up process, which is the main tool in the desingularization of an algebraic variety. The following theorem is an analytic version of Hironaka’s theorem [9] used by Atiyah [10].

**Theorem** **1**(Desingularization [9])**.** *Let f be an analytic function in a neighborhood of $w^{*}=\left(w_{1}^{*},\cdots ,w_{d}^{*}\right)\in R^{d}$ with $f(w^{*})=0$. There exists an open set $U∋w^{*}$, an analytic manifold M, and a proper analytic map μ from M to U such that: (1) $\mu :M-E\to U-f^{-1}\left(0\right)$ is an isomorphism, where $E=\mu^{-1}\left(f^{-1}\left(0\right)\right)$, and: (2) for each u∈M, there is a local analytic coordinate system $\left(u_{1},\cdots ,u_{n}\right)$ such that $f\left(\mu \left(u\right)\right)=\pm u_{1}^{s_{1}}u_{2}^{s_{2}}\cdots u_{n}^{s_{n}}$, where $s_{1},\cdots ,s_{n}$ are non-negative integers.*

The theorem establishes the existence of the desingularization map; however, generally, it is still difficult to obtain such maps for Kullback functions because the singularities of these functions are very complicated. From learning coefficient λ and its order θ, value ν is obtained theoretically as follows: Let ξ(u) be an empirical process defined on the manifold obtained by a resolution of singularities and $\sum_{u^{*}}$ denote the sum of local coordinates that attain the minimum λ and maximum θ. We then have:
(1)$$\nu =\frac{1}{2}E_{\xi }\frac{\int_{0}^{\infty }dt\sum_{u^{*}}\int du\xi \left(u\right)t^{\lambda -1/2}e^{-\beta t+\beta \sqrt{t}\xi \left(u\right)}}{\int_{0}^{\infty }dt\sum_{u^{*}}\int dut^{\lambda -1/2}e^{-\beta t+\beta \sqrt{t}\xi \left(u\right)}},$$where ξ(u) is a random variable of a Gaussian process with mean zero, and covariance $E_{\xi }\left[\xi \left(w\right)\xi \left(u\right)\right]=E_{X}\left[a\left(x,w\right)a\left(x,u\right)\right]$ for the analytic function a(x,w) obtained by the resolution of singularities using log(q(x)/p(x|w)).

Our purpose in this paper is to obtain λ. In recent studies, we determined the learning coefficients for reduced rank regression [11], the three-layered neural network with one input unit and one output unit [12,13], normal mixture models with a dimension of one [14], and the restricted Boltzmann machine [15]. Additionally, Rusakov and Geiger [16,17] and Zwiernik [18], respectively, obtained the learning coefficients for naive Bayesian networks and directed tree models with hidden variables. Drton et al. [19] considered these coefficients of the Gaussian latent tree and forest models.

The papers [20,21] derived bounds on the learning coefficients for Vandermonde matrix-type singularities and explicit values under some conditions.

The remainder of this paper is structured as follows: In Section 2, we introduce log canonical thresholds in algebraic geometry. In Section 3, we summarize key theorems for obtaining learning coefficients for learning theory. In Section 4, we present our main results. We consider the log canonical thresholds of Vandermonde matrix-type singularities (Definition 3). We present our conclusions in Section 5.

## 2. Log Canonical Threshold

**Definition** **1.**
*Let f be an analytic function in neighborhood U of $w^{*}$. Let ψ be a $C^{\infty }$ function with a compact support. Define log canonical threshold $\lambda_{w^{*}}\left(f,\psi \right)$ as the largest pole of $\int_{U}{\left|f\right|}^{2z}\psi dw$ over C or $\int_{U}{\left|f\right|}^{z}\psi dw$ over R. Additionally, define $\theta_{w^{*}}\left(f,\psi \right)$ by its order. If $\psi (w^{*})\neq 0$, then we define $\lambda_{w^{*}}\left(f\right)=\lambda_{w^{*}}\left(f,\psi \right)$ and $\theta_{w^{*}}\left(f\right)=\theta_{w^{*}}\left(f,\psi \right)$ because the log canonical threshold and its order are independent of ψ.*


Applying Hironaka’s Theorem 1 to function f(w), we have the proper analytic map μ from manifold *M* to neighborhood *U* of $w^{*}$ that satisfies Hironaka’s Theorems (1) and (2). Then, integration $\int_{U}{\left|f\right|}^{z}\psi \left(w\right)dw$ is equal to $\int_{M}{\left|f\circ \mu \right|}^{z}\psi \left(\mu \left(u\right)\right)\mu^{\prime }\left(u\right)du$, which is the sum of $\int_{U_{M}}|u_{1}^{s_{1}}u_{2}^{s_{2}}\cdots u_{d}^{s_{d}}{|}^{z}\psi \left(\mu \left(u\right)\right)\left|\mu^{\prime }\left(u\right)\right|du$, where $\left(u_{1},\cdots ,u_{d}\right)$ is a local analytic coordinate system on $U_{M}\subset M$. Therefore, the poles can be obtained. Note that for each $w^{*}$ with $f(w^{*})\neq 0$, there exists neighborhood *U* such that f(w)≠0 for all w∈U. Thus, $\int_{U}{\left|f\right|}^{z}\psi \left(w\right)dw$ has no poles. The learning coefficient is the log canonical threshold of the Kullback function (relative entropy) over the real field.

## 3. Main Theorems

In this section, several theorems are introduced for obtaining real log canonical thresholds over the real field. Theorem 2 (the method for determining the deepest singular point), Theorem 3 (the method to add variables), and Theorem 4 (the rational blowing-up method) are very helpful for obtaining the log canonical threshold. These theorems over the real field are useful for reducing the number of changes of coordinates via blow-ups.

We denote constants, such as $a^{*}$, $b^{*}$, and $w^{*}$, by suffix ∗. Define the norm of a matrix $C=(c_{ij})$ as $\left||C|\right|=\sqrt{\sum_{i,j}{\left|c_{ij}\right|}^{2}}$. Set $N_{+0}=N\cup \left\{0\right\}$.

**Lemma** **1**([14,22,23])**.** *Let U be a neighborhood of $w^{*}\in R^{d}$. Consider the ring of analytic functions on U. Let J be the ideal generated by $f_{1},\ldots ,f_{n}$, which are analytic functions defined on U. (1) If $g_{1}^{2}+\ldots +g_{m}^{2}\leq f_{1}^{2}+\cdots +f_{n}^{2}$, then $\lambda_{w^{*}}\left(g_{1}^{2}+\cdots +g_{m}^{2}\right)\leq \lambda_{w^{*}}\left(f_{1}^{2}+\cdots +f_{n}^{2}\right).$*
*(2) If $g_{1},\ldots ,g_{m}\in J$, then $\lambda_{w^{*}}\left(g_{1}^{2}+\cdots +g_{m}^{2}\right)\leq \lambda_{w^{*}}\left(f_{1}^{2}+\cdots +f_{n}^{2}\right).$ In particular, if $g_{1},\ldots ,g_{m}$ generate ideal J, then $\lambda_{w^{*}}\left(f_{1}^{2}+\cdots +f_{n}^{2}\right)=\lambda_{w^{*}}\left(g_{1}^{2}+\cdots +g_{m}^{2}\right).$*


The following lemma is also used in the proofs.

**Lemma** **2**([15])**.** *Let $J,J^{\prime }$ be the ideals generated by $f_{1}\left(w\right),\ldots ,f_{n}\left(w\right)$ and $g_{1}\left(w^{\prime }\right),\ldots ,g_{m}\left(w^{\prime }\right)$, respectively. If w and $w^{\prime }$ are different variables, then:*
$$\lambda_{\left(w^{*},w^{\prime *}\right)}\left(f_{1}^{2}+\cdots +f_{n}^{2}+g_{1}^{2}+\cdots +g_{m}^{2}\right)=\lambda_{w^{*}}\left(f_{1}^{2}+\cdots +f_{n}^{2}\right)+\lambda_{w^{\prime *}}\left(g_{1}^{2}+\cdots +g_{m}^{2}\right).$$

**Theorem** **2**(Method for determining the deepest singular point [21])**.** *Let $f_{1}\left(w_{1},\ldots ,w_{d}\right)$, …, $f_{m}\left(w_{1},\ldots ,w_{d}\right)$ be homogeneous functions of $w_{1},\cdots ,w_{j}$(j≤d). Furthermore, let ψ be a $C^{\infty }$ function such that $\psi (0,\cdots ,0,w_{j+1}^{*},\cdots ,w_{d}^{*})\geq \psi (w_{1}^{*},\cdots ,w_{d}^{*})$ and $\psi_{w}$ is a homogeneous function of $w_{1},\cdots ,w_{j}$ in a small neighborhood of $\left(0,\cdots ,0,w_{j+1}^{*},\cdots ,w_{d}^{*}\right)$. Then, we have:*
$$\lambda_{\left(0,\cdots ,0,w_{j+1}^{*},\cdots ,w_{d}^{*}\right)}\left(f_{1}^{2}+\cdots +f_{m}^{2},\psi \right)\leq \lambda_{\left(w_{1}^{*},\cdots ,w_{j}^{*},w_{j+1}^{*},\cdots ,w_{d}^{*}\right)}\left(f_{1}^{2}+\cdots +f_{m}^{2},\psi \right)$$

**Theorem** **3**(Method to add variables [21])**.** *Let $f_{1}\left(w_{1},\ldots ,w_{d}\right)$, …, $f_{m}\left(w_{1},\ldots ,w_{d}\right)$ be homogeneous functions of $w_{1},\cdots ,w_{d}$. Set $f_{1}^{\prime }\left(w_{2},\ldots ,w_{d}\right)=f_{1}\left(1,w_{2},\ldots ,w_{d}\right)$, …, $f_{m}^{\prime }\left(w_{2},\ldots ,w_{d}\right)=f_{m}\left(1,w_{2},\ldots ,w_{d}\right)$. If $w_{1}^{*}\neq 0$, then we have:*
$$\lambda_{\left(w_{1}^{*},\cdots ,w_{d}^{*}\right)}\left(f_{1}^{2}+\cdots +f_{m}^{2}\right)=\lambda_{\left(w_{2}^{*}/w_{1}^{*},\cdots ,w_{d}^{*}/w_{1}^{*}\right)}\left(f_{1}^{\prime 2}+\cdots +f_{m}^{\prime 2}\right).$$

Resolutions of singularities are obtained by constructing the blow-up along the smooth submanifold. In this paper, we use the blow-up method along some singular varieties as explained below for obtaining log canonical thresholds.

**Theorem** **4** (Rational blow-up process)**.**
*Let m≤d and $\left(\alpha_{1},\ldots ,\alpha_{m}\right)\in R^{m},\alpha_{1},\ldots ,\alpha_{m}>0$. Consider the set $U=\{w=\left(w_{1},\ldots ,w_{d}\right)\in R^{d}\;|\;0\leq w_{1},\ldots ,w_{d}\leq 1\}$.*

*We set $g_{i}\left(u_{i}\right)=\left(g_{1i}\left(u_{i}\right),\ldots ,g_{di}\left(u_{i}\right)\right)$ for $g_{ii}\left(u_{i}\right)=u_{ii}$, $g_{ji}\left(u_{i}\right)=u_{ji}u_{ii}^{\alpha_{i}/\alpha_{j}}\left(1\leq j\leq m,j\neq i\right)$, $g_{ji}\left(u_{i}\right)=u_{ji}\left(1\leq j\leq d\right)$.*

*Let f, ψ be analytic functions defined on U and $f_{i}\left(u_{1i},\cdots ,u_{di}\right)=f\left(g_{i}\left(u_{i}\right)\right)$. Then, we have:*
$$\min_{1\leq i\leq m}\left\{\lambda_{0}\left(f_{i}\left(u_{i}\right),u_{ii}^{\sum_{j=1}^{m}\alpha_{i}/\alpha_{j}-1}\psi \left(g_{i}\left(u_{i}\right)\right)\right)\right\}=\lambda_{0}\left(f,\psi \right).$$


**Proof.** The proof of this theorem uses a resolution of singularities along a smooth submanifold.Set $w_{i}^{\prime }=w_{i}^{1/\alpha_{i}}$, and construct the blow-up along the submanifold $\left\{w_{1}^{\prime }=\cdots =w_{m}^{\prime }=0\right\}$. Then, we have for 1≤i≤m, $w_{i}^{\prime }=v$, $w_{j}^{\prime }=vw_{j}^{\prime \prime }$(1≤j≤m,j≠i), and $w_{j}^{\prime }=w_{j}^{\prime \prime }$(m+1≤j≤d).Consider the set $U_{i}=\left\{u_{i}=\left(u_{1i},\ldots ,u_{di}\right)\in R^{d}\;|\;0\leq u_{1i},\ldots ,u_{di}\leq 1\right\}$ for 1≤i≤m.Set $u_{ii}=v^{\alpha_{i}},u_{ji}={w_{j}^{\prime \prime }}^{\alpha_{j}}\left(j\neq i\right)$, then we have $w_{i}=u_{ii}$, $w_{j}=u_{ii}^{\alpha_{j}/\alpha_{i}}u_{ji}$(1≤j≤m,j≠i), and $w_{j}=u_{ji}\left(m+1\leq j\leq d\right)$.The Jacobian $\partial w/\partial u_{i}$ is $\prod_{1\leq j\leq m,j\neq i}u_{ii}^{\alpha_{j}/\alpha_{i}}$. □

The theorem is simple; however, it is a useful tool for obtaining log canonical thresholds.

## 4. Main Results

In this section, we apply the theorems in Section 3 to Vandermonde matrix-type singularities, which are generic and essential in learning theory. Their associated log canonical thresholds provide the learning coefficients of, for example, three-layered neural networks in Section 4.1, normal mixture models in Section 4.2, and mixtures of binomial distributions [24].

### 4.1. Three-Layered Neural Network

Consider the three-layered neural network with *N* input units, *H* hidden units, and *M* output units, which is trained for estimating the true distribution with *r* hidden units. Denote an input value by $z=\left(z_{j}\right)\in R^{N}$ with a probability density function q(z). Then, an output value $y=\left(y_{k}\right)\in R^{M}$ of the three layered neural network is given by $y_{k}=f_{k}\left(z,w\right)+\left(\mathrm{noise}\right),$ where $w={\left\{a_{ki},b_{ij}\right\}}_{1\leq i\leq H}$ and:$$f_{k}\left(z,w\right)=\sum_{i=1}^{H}a_{ki}\tanh\left(\sum_{j=1}^{N}b_{ij}z_{j}\right).$$Consider a statistical model:
$$p\left(y|z,w\right)=\frac{1}{{\left(2\pi \right)}^{M/2}}\exp(-\frac{1}{2}||y-{f\left(z,w\right)||}^{2}),$$and p(z,y|w)=p(y|z,w)q(z). Assume that the true distribution:$$p\left(y|z,w_{t}^{*}\right)=\frac{1}{{\left(2\pi \right)}^{M/2}}\exp(-\frac{1}{2}||y-f\left(z,w_{t}^{*}\right){\left|\right|}^{2}),$$is included in the learning model, where $w_{t}^{*}={\left\{a_{k,H+i}^{*},b_{H+i,j}^{*}\right\}}_{1\leq i\leq r}$ and $f_{k}\left(z,w_{t}^{*}\right)=\sum_{i=1}^{r}\left(-a_{k,H+i}^{*}\right)\tanh\left(\sum_{j=1}^{N}b_{H+i,j}^{*}x_{j}\right).$

We have:$$p\left(z,y|w\right)=\frac{1}{{\left(2\pi \right)}^{M/2}}\exp(-\frac{1}{2}||y-{f\left(z,w\right)||}^{2})q\left(z\right),$$and the notation (z,y) for the three-layered neural network corresponds to *x* in Section 1 and Section 2.

### 4.2. Normal Mixture Models

We consider a normal mixture model [14] with identity matrix variances:$$p\left(x|w\right)=\frac{1}{{\left(2\pi \right)}^{N/2}}\sum_{i=1}^{H}a_{i}\exp\left(-\frac{\sum_{j=1}^{N}{\left(z_{j}-b_{ij}\right)}^{2}}{2}\right),$$where $w={\left\{a_{i},b_{ij}\right\}}_{1\leq i\leq H}$ and $\sum_{i=1}^{H}a_{i}=1$, $a_{i}\geq 0$.

Set the true distribution by:$$p\left(x|w_{t}^{*}\right)=\frac{1}{{\left(2\pi \right)}^{N/2}}\sum_{i=H+1}^{H+r}\left(-a_{i}^{*}\right)\exp\left(-\frac{\sum_{j=1}^{N}{\left(z_{j}-b_{ij}^{*}\right)}^{2}}{2}\right),$$where $w_{t}^{*}={\left\{a_{i}^{*},b_{ij}^{*}\right\}}_{H+1\leq i\leq H+r}$ and $\sum_{i=H+1}^{H+r}a_{i}^{*}=-1$, $a_{i}^{*}<0$. (In order to simplify the following, we use the values $a_{i}^{*}<0$, not $a_{i}^{*}>0$.)

### 4.3. Vandermonde Matrix-Type Singularities

**Definition** **2.**
*Fix Q∈N. Define: ${\left[b_{1}^{*},b_{2}^{*},\cdots ,b_{N}^{*}\right]}_{Q}=\gamma_{i}\left(0,\cdots ,0,b_{i}^{*},\cdots ,b_{N}^{*}\right)$ if $b_{1}^{*}=\cdots =b_{i-1}^{*}=0$, $b_{i}^{*}\neq 0$, and $\gamma_{i}=\left\{\begin{array}{cc} 1 & \mathrm{if}\;Q\;\mathrm{is}\;\mathrm{odd},\\ \mathrm{sign}(b_{i}^{*}) & \mathrm{if}\;Q\;\mathrm{is}\;\mathrm{even}. \end{array}\right.$*


For simplicity, we use the notation $w={\left\{a_{ki},b_{ij}\right\}}_{1\leq i\leq H}$ instead of $w={\left\{a_{ki},b_{ij}\right\}}_{1\leq k\leq M,1\leq i\leq H,1\leq j\leq N}$ because we always have 1≤k≤M and 1≤j≤N in this section.

**Definition** **3.**
*Fix Q∈N and $m\in N_{+0}$.*

*Let $A=\left(\begin{array}{cccccc} a_{11} & \cdots & a_{1H} & a_{1,H+1}^{*} & \ldots & a_{1,H+r}^{*}\\ a_{21} & \cdots & a_{2H} & a_{2,H+1}^{*} & \ldots & a_{2,H+r}^{*}\\ & \vdots & & & \vdots & \\ a_{M1} & \cdots & a_{MH} & a_{M,H+1}^{*} & \ldots & a_{M,H+r}^{*} \end{array}\right)$,*
$$B_{I}={\left(\prod_{j=1}^{N}b_{1j}^{\ell_{j}},\prod_{j=1}^{N}b_{2j}^{\ell_{j}},\cdots ,\prod_{j=1}^{N}b_{Hj}^{\ell_{j}},\prod_{j=1}^{N}{b_{H+1,j}^{*}}^{\ell_{j}},\cdots ,\prod_{j=1}^{N}{b_{H+r,j}^{*}}^{\ell_{j}}\right)}^{t},$$
*for $I=\left(\ell_{1},\ldots ,\ell_{N}\right)\in {N_{+0}}^{N}$, and:*
$$B={\left(B_{I}\right)}_{\ell_{1}+\cdots +\ell_{N}=Qn+m,n\geq 0}=\left(B_{\left(m,0,\cdots ,0\right)},B_{\left(m-1,1,\cdots ,0\right)},\cdots ,B_{\left(0,0,\cdots ,m\right)},B_{\left(m+Q,0,\cdots ,0\right)},\cdots \right)$$
*(t denotes the transpose).*

*$a_{ki}$ and $b_{ij}$(1≤k≤M,1≤i≤H,1≤j≤N) are the variables in a neighborhood of $a_{ki}^{*}$ and $b_{ij}^{*}$, where $a_{ki}^{*}$ and $b_{ij}^{*}$ are fixed constants.*

*Let J be the ideal generated by the elements of AB.*

*We call singularities of J Vandermonde matrix-type singularities.*

*To simplify, we usually assume that:*
$${\left(a_{1,H+j}^{*},a_{2,H+j}^{*},\cdots ,a_{M,H+j}^{*}\right)}^{t}\neq 0,\left(b_{H+j,1}^{*},b_{H+j,2}^{*},\cdots ,b_{H+j,N}^{*}\right)\neq 0,$$
*for 1≤j≤r and:*
$${\left[b_{H+j,1}^{*},b_{H+j,2}^{*},\cdots ,b_{H+j,N}^{*}\right]}_{Q}\neq {\left[b_{H+j^{\prime },1}^{*},b_{H+j^{\prime },2}^{*},\cdots ,b_{H+j^{\prime },N}^{*}\right]}_{Q},$$
*for $j\neq j^{\prime }$.*


**Example** **1.**
*If m=N=M=r=1, Q=2, H=3, then we have $A=\left(\begin{array}{cccc} a_{11} & a_{12} & a_{13} & a_{14}^{*} \end{array}\right)$, $B=\left(\begin{array}{cccc} b_{11} & b_{11}^{3} & b_{11}^{5} & b_{11}^{7}\\ b_{21} & b_{21}^{3} & b_{11}^{5} & b_{21}^{7}\\ b_{31} & b_{31}^{3} & b_{11}^{5} & b_{31}^{7}\\ b_{41}^{*} & {b_{41}^{*}}^{3} & {b_{41}^{*}}^{5} & {b_{41}^{*}}^{7} \end{array}\right)$. These matrices A,B correspond to the three-layered neural network:*
$$p\left(y|z,w\right)=\frac{1}{{\left(2\pi \right)}^{1/2}}\exp\left(-\frac{1}{2}{\left(y-a_{11}\tanh\left(b_{11}z\right)-a_{12}\tanh\left(b_{21}z\right)-a_{13}\tanh\left(b_{31}z\right)\right)}^{2}\right),$$
*and the true distribution:*
$$p\left(y|x,w_{t}^{*}\right)=\frac{1}{{\left(2\pi \right)}^{1/2}}\exp\left(-\frac{1}{2}{\left(y+a_{14}^{*}\tanh\left(b_{41}^{*}z\right)\right)}^{2}\right).$$


**Example** **2.**
*If Q=r=M=1, H=2,N=2, then we have $A=\left(\begin{array}{ccc} a_{11} & a_{12} & a_{13}^{*} \end{array}\right)$, $B=\left(\begin{array}{ccccccccc} b_{11} & b_{12} & b_{11}^{2} & b_{11}b_{12} & b_{12}^{2} & b_{11}^{3} & b_{11}b_{12}^{2} & b_{11}^{2}b_{12} & b_{12}^{3}\\ b_{21} & b_{22} & b_{21}^{2} & b_{21}b_{22} & b_{22}^{2} & b_{21}^{3} & b_{21}b_{22}^{2} & b_{21}^{2}b_{22} & b_{22}^{3}\\ b_{31}^{*} & b_{32}^{*} & {b_{31}^{*}}^{2} & b_{31}^{*}b_{32}^{*} & {b_{32}^{*}}^{2} & {b_{31}^{*}}^{3} & b_{31}^{*}{b_{32}^{*}}^{2} & {b_{31}^{*}}^{2}b_{32}^{*} & {b_{32}^{*}}^{3} \end{array}\right).$*

*If $a_{13}^{*}=-1$, these matrices A,B correspond to a normal mixture model with identity matrix variances:*
$$p\left(x|w\right)=\frac{a_{11}}{2\pi }\exp\left(-\frac{{\left(z_{1}-b_{11}\right)}^{2}+{\left(z_{2}-b_{12}\right)}^{2}}{2}\right)+\frac{a_{12}}{2\pi }\exp\left(-\frac{{\left(z_{1}-b_{21}\right)}^{2}+{\left(z_{2}-b_{22}\right)}^{2}}{2}\right),$$
*$\sum_{i=1}^{2}a_{1i}=1$, $a_{1i}\geq 0$, and the true distribution is:*
$$p\left(x|w_{t}^{*}\right)=\frac{1}{2\pi }\left(-a_{13}^{*}\right)\exp\left(-\frac{{\left(z_{1}-b_{31}^{*}\right)}^{2}+{\left(z_{2}-b_{32}^{*}\right)}^{2}}{2}\right),-a_{13}^{*}=1.$$


In this paper, we denote:$$\begin{array}{c} A_{M,H}=\left(\begin{array}{cccc} a_{11} & a_{12} & \cdots & a_{1H}\\ a_{21} & a_{22} & \cdots & a_{2H}\\ & & \vdots & \\ a_{M1} & a_{M2} & \cdots & a_{MH} \end{array}\right),B_{H,N,I}=\left(\begin{array}{c} \prod_{j=1}^{N}{b_{1j}}^{\ell_{j}}\\ \prod_{j=1}^{N}{b_{2j}}^{\ell_{j}}\\ \vdots \\ \prod_{j=1}^{N}{b_{Hj}}^{\ell_{j}} \end{array}\right)and: \end{array}$$$B_{H,N}^{\left(Q\right)}={\left(B_{H,N,I}\right)}_{\ell_{1}+\ldots +\ell_{N}=Qn+1,0\leq n\leq H-1}$.

Furthermore, we denote: $a^{*}=\left(\begin{array}{c} a_{1,H+1}^{*}\\ \vdots \\ a_{M,H+1}^{*} \end{array}\right)$ and:$$\begin{array}{c} \left(A_{M,H},a^{*}\right)=\left(\begin{array}{ccccc} a_{11} & a_{12} & \cdots & a_{1H} & a_{1,H+1}^{*}\\ a_{21} & a_{22} & \cdots & a_{2H} & a_{2,H+1}^{*}\\ & & \vdots & & \\ a_{M1} & a_{M2} & \cdots & a_{MH} & a_{M,H+1}^{*} \end{array}\right) \end{array}$$

**Theorem** **5**([14])**.** *Consider sufficiently small neighborhood U of:*
$$w^{*}={\left\{a_{ki}^{*},b_{ij}^{*}\right\}}_{1\leq i\leq H}$$*and variables $w={\left\{a_{ki},b_{ij}\right\}}_{1\leq i\leq H}$ in set U. Set $\left(b_{01}^{**},b_{02}^{**},\cdots ,b_{0N}^{**}\right)=\left(0,\ldots ,0\right)$. Let each $\left(b_{11}^{**},b_{12}^{**},\cdots ,b_{1N}^{**}\right)$, …, $\left(b_{r^{\prime }1}^{**},b_{r^{\prime }2}^{**},\cdots ,b_{r^{\prime }N}^{**}\right)$ be a different real vector in:*
$${\left[b_{i1}^{*},b_{i2}^{*},\cdots ,b_{iN}^{*}\right]}_{Q}\neq 0,for\;i=1,\ldots ,H+r;$$*that is,*
$$\{\left(b_{11}^{**},\cdots ,b_{1N}^{**}\right),\ldots ,\left(b_{r^{\prime }1}^{**},\cdots ,b_{r^{\prime }N}^{**}\right)\;;\;{\left[b_{i1}^{*},\cdots ,b_{iN}^{*}\right]}_{Q}\neq 0,i=1,\ldots ,H+r\}.$$*Then, $r^{\prime }$ is uniquely determined, and $r^{\prime }\geq r$ by the assumption in Definition 3. Set $\left(b_{i1}^{**},\cdots ,b_{iN}^{**}\right)={\left[b_{H+i,1}^{*},\cdots ,b_{H+i,N}^{*}\right]}_{Q},$ for 1≤i≤r. Assume that ${\left[b_{i1}^{*},\cdots ,b_{iN}^{*}\right]}_{Q}=\left\{\begin{array}{cc} 0, & 1\leq i\leq H_{0}\\ (b_{11}^{**},\cdots ,b_{1N}^{**}), & H_{0}+1\leq i\leq H_{0}+H_{1},\\ (b_{21}^{**},\cdots ,b_{2N}^{**}), & H_{0}+H_{1}+1\leq i\leq H_{0}+H_{1}+H_{2},\\ \vdots \\ (b_{r^{\prime }1}^{**},\cdots ,b_{r^{\prime }N}^{**}), & H_{0}+\cdots +H_{r^{\prime }-1}+1\leq i\leq H_{0}+\cdots +H_{r^{\prime }}, \end{array}\right.$*
*and $H_{0}+\cdots +H_{r^{\prime }}=H$. We then have:*
$$\lambda_{w^{*}}{\left(||AB|\right|}^{2})=\frac{Mr^{\prime }}{2}+\lambda_{{w_{1}^{\left(0\right)}}^{*}}\left(|\right|A_{M,H_{0}}B_{H_{0},N}^{\left(Q\right)}{\left|\right|}^{2})$$
$$+\sum_{\alpha =1}^{r}\lambda_{{w_{1}^{\left(\alpha \right)}}^{*}}\left(|\right|\left(A_{M,H_{\alpha }-1},{a^{\left(\alpha \right)}}^{*}\right)B_{H_{\alpha },N}^{\left(1\right)}{\left|\right|}^{2})+\sum_{\alpha =r+1}^{r^{\prime }}\lambda_{{w_{1}^{\left(\alpha \right)}}^{*}}\left(|\right|A_{M,H_{\alpha }-1}B_{H_{\alpha }-1,N}^{\left(1\right)}{\left|\right|}^{2}),$$
*where ${w_{1}^{\left(0\right)}}^{*}={\left\{a_{k,i}^{*},0\right\}}_{1\leq i\leq H_{\alpha }},$*

*${w_{1}^{\left(\alpha \right)}}^{*}={\left\{a_{k,H_{0}+\cdots +H_{\alpha -1}+i}^{*},0\right\}}_{2\leq i\leq H_{\alpha }}$ and ${a^{\left(\alpha \right)}}^{*}=\left(\begin{array}{c} a_{1,H+\alpha }^{*}\\ \vdots \\ a_{M,H+\alpha }^{*} \end{array}\right)$ for α≥1.*


**Theorem** **6.**
*We use the same notation as in Theorem 5. Set $\lambda =\lambda_{0}\left(|\right|A_{M,H}B_{H,N}^{\left(Q\right)}{\left|\right|}^{2})$.*

*We have the following:*
*1.* 
*H=1. $\lambda =\min\{\frac{M}{2},\frac{N}{2}\}$.*
*2.* 
*H=2. $\lambda =\min\{\frac{\beta N+(2-\beta )M}{2}$, β=0,1,2, $\frac{2N+Q(N-1+M)}{2Q+2}\}$.*
*3.* 
*H=3.*

*$\lambda =\min\{\frac{\beta N+(3-\beta )M}{2}$, β=0,1,2,3,*

*$\frac{\beta N+(3-\beta )M+Q(\alpha (N+\alpha -\beta )+(3-\alpha )M)}{2(Q+1)},$α=1,⋯,β−1, β=2,3,*

$\frac{3N+Q(3N-3+3M)}{2(2Q+1)}\}$
*4.* 
*H=4.*

*$\lambda =\min\{\frac{\beta N+(4-\beta )M}{2}$, β=0,1,2,3,4,*

*$\frac{\beta N+(4-\beta )M+Q(\alpha (N+\alpha -\beta )+(4-\alpha )M)}{2(Q+1)},$α=1,⋯,β−1, β=2,3,4,*

*$\frac{4N+Q(\alpha N-\alpha -1+(8-\alpha )M)}{2(2Q+1)},$α=2,3,4, $\frac{4N+Q(5N-5+3M)}{2(2Q+1)},$*

*$\frac{3N+M+Q(\alpha N-\alpha +(8-\alpha )M)}{2(2Q+1)},$α=2,3,*

$\frac{4N+Q(6N-6+6M)}{2(3Q+1)}\}.$



Its proof appears in Appendix A.

In paper [22], we had exact values for N=1:$$\lambda_{0}\left(|\right|A_{M,H}B_{H,1}^{\left(Q\right)}{\left|\right|}^{2})=\frac{MQk(k+1)+2H}{4(1+kQ)}$$where: k=max{i∈Z:2H≥M(i(i−1)Q+2i)}, and we had:$$\theta =\left\{\begin{array}{cc} 1, & \mathrm{if}\;2H>M(k(k-1)Q+2k)\\ 2, & \mathrm{if}\;2H=M(k(k-1)Q+2k) \end{array}\right.$$

## 5. Conclusions

In this paper, we proposed a new method of “rational blowing-up” (Theorem 4), and we applied the method to Vandermonde matrix-type singularities and demonstrated its effectiveness. Theorem 6 determines the explicit values of log canonical thresholds for H=1,2,3,4. Our future research aim is to improve our methods and obtain explicit values for the general model.

These theoretical values introduce a mathematical measure of preciseness for numerical calculations in information criteria in Section 1. Furthermore, our theoretical results will be helpful in numerical experiments such as the Markov chain Monte Carlo (MCMC). In the papers [25,26], the mathematical foundation for analyzing and developing the precision of the MCMC method was constructed by using the theoretical values of marginal likelihoods.

We will also consider these applications in the future.

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
