# Peer review of "Learning Coefficient of Vandermonde Matrix-Type Singularities in Model Selection"

_entropy, 2019, doi:10.3390/e21060561_

Round 1

Reviewer 1 Report

This is an interesting paper on deep issues on statistical inference in the Bayesian paradigm.  Given the short time for a review I am unable to check proofs and mathematical detail but I have the following comments:

1.  Motivation of the paper: The best motivation for computing learning coefficients is the fact that they are used explicitly in the model selection method (sBIC) of Drton & Plummer.  The coefficients are only implicit in the derivation of other criteria such as WAIC but are not needed to evaluate WAIC in practice.  As such I find the abstract somewhat misleading.  I would suggest that the author avoids mentioning specific model selection techniques in the abstract and then discusses (via citation) the importance of learning coefficients in sBIC and in the theory that other criteria such as WAIC build on.

2.  Title: I would suggest "... in model selection" (i.e., omit the word "methods")

3.  Introduction: I would rephrase as "... WAIC [1-7] and cross-validation in Bayesian model selection."

4.  The text is not broken into paragraphs.  For example, in the introduction I would start a new paragraph when saying "Let q(x) be a true probabilit density function..."  Similarly the rest of the text in the introduction should be more broken up into paragraphs.

5.  page 1: Def of E_w[f(w)]: f(x) -> f(w)

6.  page 2: in def of B_t: more logical to have x instead of x_i?

7.  page 2: The information criterion in the WAIC -> The WAIC; and in the cross-validation loss -> and the cross-validation loss

8.  page 3: For singular models, the set W_0 is typically not a singleton set.  Nevertheless, WAIC and cross-validation can estimate... of the true ...function.  [New paragraph]

9.  page 3: Our purpose is to show that these methods are effective [cut words]

10. Theorem 1: Let f be an analytic function ... There exists an open set ..., an analytic manifold..., and a proper...

11. after the Theorem 1: The theorem establishes the existence...  it is still difficult to obtain such maps...

12. Gaussian process with mean zero and variance two.  What is the covariance function?

13. page 3: References on learning coefficients.  The author should also add:

Drton, Mathias; Lin, Shaowei; Weihs, Luca; Zwiernik, Piotr. Marginal likelihood and model selection for Gaussian latent tree and forest models. Bernoulli 23 (2017), no. 2, 1202--1232.  doi:10.3150/15-BEJ775. https://projecteuclid.org/euclid.bj/1486177397

14. page 4, Section 3:

over the real field

useful for reducing the number of changes of coordinates via blow-ups.

Define the norm of a matrix

15. Lemma 1: Please indicate in which ring f_1,...f_n generate the ideal J

16. Theorems 2 and 3: I did not understand the notation "with degree n_i" What is i referring to?  where do these degrees appear?

17. Theorem 2: There is psi_(0,...  these should not be in subscript.

18. Theorem 4: The text is confusing.  Is this a result from the literature or a new result.  If old please add a reference; if new then please give the proof (even if easy).  The sets U_i are defined in the theorem but are not used in the writing anywhere, which is confusing for a reader.

19. Section 4:

These log canonical ->  Their assocated log canonical ...

Delete "which are widely used as effective learning methods"

20.  Def 2:  |b_i^*|/b_i* ;  you could write sign(b_i^*)

21.  B_H,N,I : The notation seems confusing.  What is I?  where does ell_j come from?  Similar for B_H,N^(Q)

22.  Theorem 5: Strictly speaking the matrices A and B have not been defined.

Author Response

Thank you very much for your fruitful comments.
I  appreciate it very much.

The attached file is my revised paper

Comments and Suggestions
for Authors
This is an interesting paper on deep issues on statistical inference in the
Bayesian paradigm. Given the short time for a review I am unable to check
proofs and mathematical detail but I have the following comments:

C1.
 Motivation of the paper: The best motivation for computing learning
coefficients is the fact that they are used explicitly in the model selection
method (sBIC) of Drton & Plummer. The coefficients are only implicit in the
derivation of other criteria such as WAIC but are not needed to evaluate
WAIC in practice. As such I find the abstract somewhat misleading. I would
suggest that the author avoids mentioning specific model selection
techniques in the abstract and then discusses (via citation) the importance of
learning coefficients in sBIC and in the theory that other criteria such as
WAIC build on.

A1.

We have added the followings in Introduction:
" This  coefficient is not needed to evaluate
in the WAIC and cross-validation in practice, while
the learning coefficients from our recent results have been used very effectively by Drton and Plummer~\cite{Drton}
for model selection using a method called the
``singular Bayesian information criterion (sBIC)''.
The method sBIC even works using bounds of learning coefficients."

C2. Title: I would suggest "... in model selection" (i.e., omit the word
"methods")

A2.  We have changed these.

C3. Introduction: I would rephrase as "... WAIC [1-7] and cross-validation in
Bayesian model selection."

A3. We have changed these.

C4. The text is not broken into paragraphs. For example, in the introduction I
would start a new paragraph when saying "Let q(x) be a true probabilit
density function..." Similarly the rest of the text in the introduction should be
more broken up into paragraphs.

A4. We have changed these.

C5. page 1: Def of E_w[f(w)]: f(x) -> f(w)

A5. We have changed these.

C6. page 2: in def of B_t: more logical to have x instead of x_i?

A6.  We have changed these.

C7. page 2: The information criterion in the WAIC -> The WAIC; and in the
cross-validation loss -> and the cross-validation loss

A7.  We have changed these.

C8. page 3: For singular models, the set W_0 is typically not a singleton set.
Nevertheless, WAIC and cross-validation can estimate... of the
true ...function. [New paragraph]

A8.  We have changed these.

C9. page 3: Our purpose is to show that these methods are effective [cut
words]

A9.  We have changed these.

C10. Theorem 1: Let f be an analytic function ... There exists an open set ..., an
analytic manifold..., and a proper...

A10.  We have changed these.

C11. after the Theorem 1: The theorem establishes the existence... it is still
difficult to obtain such maps...

A11.  We have changed these.

C12. Gaussian process with mean zero and variance two. What is the
covariance function?

A12.
We have added
"where $\xi(u)$ is a random variable of a Gaussian process with mean zero,
 and covariance $E_\xi[\xi(w)\xi(u)]=E_X[a(x, w)a(x, u)]$
for the analytic function $a(x,w)$ obtained by resolution of singularities using
$\log(q(x)/p(x|w))$. "

C13. page 3: References on learning coefficients. The author should also add:
Drton, Mathias; Lin, Shaowei; Weihs, Luca; Zwiernik, Piotr. Marginal
likelihood and model selection for Gaussian latent tree and forest models.
Bernoulli 23 (2017), no. 2, 1202--1232. doi:10.3150/15-BEJ775.
https://projecteuclid.org/euclid.bj/1486177397

A13.
We have added
"In the paper \cite{DLLP}, Drton et~al. considered these coefficients
of Gaussian latent tree and forest models."

C14. page 4, Section 3:
over the real field
useful for reducing the number of changes of coordinates via blow-ups.
Define the norm of a matrix

A14.  We have changed these.

C15. Lemma 1: Please indicate in which ring f_1,...f_n generate the ideal J

A15.  We have added
"Consider the ring of analytic functions on $U$."

C16. Theorems 2 and 3: I did not understand the notation "with degree n_i"
What is i referring to? where do these degrees appear?

A16.  We have omitted them.

C17. Theorem 2: There is psi_(0,... these should not be in subscript.

A17.  We have changed these.

C18. Theorem 4: The text is confusing. Is this a result from the literature or a
new result. If old please add a reference; if new then please give the proof
(even if easy). The sets U_i are defined in the theorem but are not used in
the writing anywhere, which is confusing for a reader.

A18.  We have added proof and changed these.

C19. Section 4:
These log canonical -> Their assocated log canonical ...
Delete "which are widely used as effective learning methods"

A19.  We have changed these.

C20. Def 2: |b_i^*|/b_i* ; you could write sign(b_i^*)

A20.  We have added the definition and changed these.

C21. B_H,N,I : The notation seems confusing. What is I? where does ell_j
come from? Similar for B_H,N^(Q)

A21.  We added the definitions.

C22. Theorem 5: Strictly speaking the matrices A and B have not been
defined.

A22.  We have changed these.

Reviewer 2 Report

I am mainly commenting about the statistical aspects of the work. I found the paper interesting and well motivated. I think the results are potentially of interest, but I feel that the paper needs a small bit extra to improve it. 

Comments:

The work is well motivated in Section 1. 

The background mathematics is well presented in Section 2-3. 

There is quite a jump at the start of Section 4 (in terms of tying the results/notation back to what came before). 

The conclusions should be expanded to tie in much better with the motivating work in Section 1.

Adding an example (however basic) of a singular model as a running example would help the reader hugely. This would greatly improve the manuscript and make it more useful to people using singular models. 

Minor comments:

The language is quite terse, eventhough generally correct.

Some sentences are very specific, but read as if they are general. For example "The purpose of the learning system is to estimate unknown true density function q(x) from xn using p(x|w) in Bayesian estimation." This would be better as "Suppose that, the ..." The extra "Suppose that," means that this is the context of this paper but not all learning systems.

Page 1. The Equations for E_w[f(w)] and V_w[f(w)] are very dependent on beta. I am suprised that the notation doesn't show this dependence. 

Page 2. The connection between T_n and B_t and G_n and B_g should be given but it only holds If beta=1.

Page 2. At the end of this page you introduce the beta term into the notation without explanation. So making the earlier edit will help. 

Author Response

Thank you very much for your fruitful comments.
I appreciate very much.
The attached file is my revised paper.

C1.
The work is well motivated in Section 1.
The background mathematics is well presented in Section 2-3.
There is quite a jump at the start of Section 4 (in terms of tying the
results/notation back to what came before).

A1.
I added \subsection{Three layered neural network}
and \subsection{Normal mixture models}
at the start of Section4:

C2.
The conclusions should be expanded to tie in much better with the motivating
work in Section 1.

A2 I added in Conclusions
"These theoretical values introduce a mathematical measure of
preciseness for numerical calculations in information criteria
in Section 1.
Also,  our theoretical results will be helpful in numerical experiments
 such as the Markov Chain Monte Carlo (MCMC).
In the paper \citep{Nagata1} and \citep{Nagata2},
 mathematical foundation for
analyzing and developing the precision of the MCMC method
is constructed
by using the theoretical values of
marginal likelihoods."

C3.
Adding an example (however basic) of a singular model as a running
example would help the reader hugely. This would greatly improve the
manuscript and make it more useful to people using singular models.

A3.
We have added two examples

Minor comments:

C4.
The language is quite terse, eventhough generally correct.
Some sentences are very specific, but read as if they are general. For
example "The purpose of the learning system is to estimate unknown true
density function q(x) from xn using p(x|w) in Bayesian estimation." This would
be better as "Suppose that, the ..." The extra "Suppose that," means that this
is the context of this paper but not all learning systems.

A4. We have changed it.

C5.
Page 1. The Equations for E_w[f(w)] and V_w[f(w)] are very dependent on
beta. I am suprised that the notation doesn't show this dependence.

A5. We have changed these.

C6.
Page 2. The connection between T_n and B_t and G_n and B_g should be
given but it only holds If beta=1.

A6. We made  some mistakes in definitions.
We have changed these.

C7.
Page 2. At the end of this page you introduce the beta term into the notation
without explanation. So making the earlier edit will help.

A7. We have changed these.

Reviewer 3 Report

The author deals with a method of calculating learning coefficients (log canonical thresholds) of specific singular models, which is an important problem of WAIC model selection framework in Bayesian inference.

It seems difficult to calculate learning coefficients of arbitrary models, therefore WAIC provides an empirical method of calculating learning coefficients with cross-validation.

Still it is beneficial to obtain learning coefficients theoretically, even if for specific models, I agree that the direction of this research is important.

However, the current version of the draft is not appropriate for publication because of its readability.

At least the following points should be reconsidered.

1. 

The introduction is nicely summarised, but the notations should be rearranged.

For example, E_w is defined as a specific expectation operator (which might be expectation under escort distribution or beta-expectation in some literature), and the author defines p(x|x^n)=E_w[p(x|w)], however, p(x|x^n) and E_w[p(x|w)] are intermingled, that is very confusing.

Also x_i and X_i are mixed, and undefined "V^beta" (which might be understood) is used.

Anyway, the author should unify the notations.

2.

Please give an explanation of "Vandermonde matrix-type singularity".

I understand it is the same with ref.[13], but the paper should be self-contained.

Define A and b and describe the target statistical model clearly.

3. 

The author should explain the difference between the main result and ref.[13], improved points, and the contribution of this paper.

Author Response

Thank you very much for your fruitful comments.

I appreciate very much.

The attached file is my revised paper.

Comments and Suggestions

for Authors

The author deals with a method of calculating learning coefficients (log

canonical thresholds) of specific singular models, which is an important

problem of WAIC model selection framework in Bayesian inference.

It seems difficult to calculate learning coefficients of arbitrary models,

therefore WAIC provides an empirical method of calculating learning

coefficients with cross-validation.

Still it is beneficial to obtain learning coefficients theoretically, even if for

specific models, I agree that the direction of this research is important.

However, the current version of the draft is not appropriate for publication

because of its readability.

At least the following points should be reconsidered.

C1.

The introduction is nicely summarised, but the notations should be

rearranged.

For example, E_w is defined as a specific expectation operator (which might

be expectation under escort distribution or beta-expectation in some

literature), and the author defines p(x|x^n)=E_w[p(x|w)], however, p(x|x^n)

and E_w[p(x|w)] are intermingled, that is very confusing.

Also x_i and X_i are mixed, and undefined "V^beta" (which might be

understood) is used.

Anyway, the author should unify the notations.

A1.  We have changed these.

C2.

Please give an explanation of "Vandermonde matrix-type singularity".

I understand it is the same with ref.[13], but the paper should be selfcontained.

Define A and b and describe the target statistical model clearly.

A2.  We have added these in Definition.

C3.

The author should explain the difference between the main result and ref.[13],

improved points, and the contribution of this paper.

A3.

We added that the subsection "Normal mixture models".

Round 2

Reviewer 1 Report

I am fine with the author's response/changes.

Reviewer 2 Report

The authors have addressed the concerns I raised. The flow of the paper is much better now.